# Analysis of 1-Aroyl-3-[3-chloro-2-methylphenyl] Thiourea Hybrids as Potent Urease Inhibitors: Synthesis, Biochemical Evaluation and Computational Approach

**DOI:** 10.3390/ijms231911646

**Published:** 2022-10-01

**Authors:** Samina Rasheed, Mubashir Aziz, Aamer Saeed, Syeda Abida Ejaz, Pervaiz Ali Channar, Seema Zargar, Qamar Abbas, Humidah Alanazi, Mumtaz Hussain, Mona Alharbi, Song Ja Kim, Tanveer A. Wani, Hussain Raza

**Affiliations:** 1Department of Chemistry, Quaid-i-Azam University, Islamabad 45320, Pakistan; 2Department of Pharmaceutical Chemistry, Faculty of Pharmacy, The Islamia University of Bahawalpur, Bahawalpur 63100, Pakistan; 3Institute of Chemistry, Shah Abdul Latif University, Khairpur 66020, Pakistan; 4Department of Biochemistry, College of Science, King Saud University, Riyadh 11451, Saudi Arabia; 5Department of Biology, College of Science, University of Bahrain, Sakhir 32038, Bahrain; 6Department of Chemistry, University of Karachi, Karachi 75270, Pakistan; 7Department of Biological Sciences, College of Natural Sciences, Kongju National University, Gongju 32588, Korea; 8Department of Pharmaceutical Chemistry, College of Pharmacy, King Saud University, P.O. Box 2457, Riyadh 11451, Saudi Arabia

**Keywords:** thiourea hybrids, jack bean urease, anti-oxidant, density functional theory (DFT), structure activity relationship (SAR)

## Abstract

Urease is an amidohydrolase enzyme that is responsible for fatal morbidities in the human body, such as catheter encrustation, encephalopathy, peptic ulcers, hepatic coma, kidney stone formation, and many others. In recent years, scientists have devoted considerable efforts to the quest for efficient urease inhibitors. In the pharmaceutical chemistry, the thiourea skeleton plays a vital role. Thus, the present work focused on the development and discovery of novel urease inhibitors and reported the synthesis of a set of 1-aroyl-3-[3-chloro-2-methylphenyl] thiourea hybrids with aliphatic and aromatic side chains **4a**–**j**. The compounds were characterized by different analytical techniques including FT-IR, ^1^H-NMR, and ^13^C-NMR, and were evaluated for in-vitro enzyme inhibitory activity against jack bean urease (JBU), where they were found to be potent anti-urease inhibitors and the inhibitory activity IC_50_ was found in the range of 0.0019 ± 0.0011 to 0.0532 ± 0.9951 μM as compared to the standard thiourea (IC_50_ = 4.7455 ± 0.0545 μM). Other studies included density functional theory (DFT), antioxidant radical scavenging assay, physicochemical properties (ADMET properties), molecular docking and molecular dynamics simulations. All compounds were found to be more active than the standard, with compound **4i** exhibiting the greatest JBU enzyme inhibition (IC_50_ value of 0.0019 ± 0.0011 µM). The kinetics of enzyme inhibition revealed that compound **4i** exhibited non-competitive inhibition with a **Ki** value of 0.0003 µM. The correlation between DFT experiments with a modest HOMO-LUMO energy gap and biological data was optimal. These recently identified urease enzyme inhibitors may serve as a starting point for future research and development.

## 1. Introduction

Urease is a member of the amidohydrolases and phosphotriesterases superfamily that is produced by plants, fungi, algae, and bacteria and is responsible for converting urea to ammonia and carbon dioxide [1,2]. The structure of urease contains two nickel atoms at the active site, which are mandatory for its activity [3]. The urease has a massive historical background as it was the first enzyme ever to be crystallized. Later, the nickel atom role was investigated thoroughly followed by a deep understanding of the urease structure of the jack bean for a better understanding of its ureolytic activity. Jack bean (CanaValia ensiformis) urease contains six subunits, each of which is made up of 840 amino acids [4,5].

The prime role of urease is to hydrolyze urea, which is the key compound responsible for the normal body physiological functions [6,7]. Excess levels of urea and its metabolites within the body result in several disorders including hepatic coma urolithiasis, hepatic encephalopathy, urinary catheter encrustation, gastric and peptic ulcers and pyelonephritis [8]. Excessive ammonia is also responsible for alkalinity of the stomach, which in return increases the gastric mucosa permeability and tears down the gastrointestinal tract (GIT) epithelium. Helicobacter pylori (HP) is involved in gastric and peptic ulcers which ultimately lead to gastric cancer, because the lowering of stomach pH facilitates bacterial growth [9,10,11]. Due to all these fatal complications, urease inhibition has been the major target for the last few years. Many anti-urease molecules have been reported, which include imidazoles [12,13,14], and benzohydroxamic acid derivatives [15] (Figure 1). But unfortunately, all these agents have adverse effects as well. Thus, we need to identify more effective anti-urease agents with low toxicity and high bioavailability.

The class of organic compound thiourea contain sulphur and have a structural resemblance to urea, i.e., the oxygen atom is replaced by the sulphur atom; they showed excellent biological applications, especially as an anti-urease activity [15,16]. In addition to this, thiourea and its derivatives have also been found to exhibit various pharmacological activities such as anti-oxidant, anti-inflammatory, anti-hypertensive, anti-epileptic, anti-cancer and anti-bacterial activity for the treatment of various co-infections and fatal diseases including renal failure, sepsis and various cancer types [17,18,19]. Considering the significance of thiourea moiety, this work is designed to synthesize 1-aroyl-3-[3-chloro-2-methylphenyl] thiourea hybrids [**4a**–**j**] as JBU inhibitors [20]. The in vitro enzyme inhibitory activity was performed. The structure activity relationships were established by incorporation of the side chain aliphatic and aromatic moieties. Molecular docking experiments were used to confirm the inhibitors’ binding conformations within the active pocket of the enzyme [21,22] and electrostatic potential surface maps derived from traditional DFT computations were employed to determine their relative strength [23]. On the basis of the results of performed activities, this paper proposes a structural model for novel potential inhibitors.

## 2. Results and Discussion

### 2.1. Chemistry

A series **4a**–**j** of novel aryl thiourea derivatives were synthesized by treating potassium thiocyanate with different acid chlorides in dry acetone refluxed for 30 min; the respective isothiocyanate intermediates formed in the reaction mixture, after cooling the reaction mixture 2-methyl-3-chloroaniline, were incorporated to afford the final products.

### 2.2. Spectroscopic Characterization

Spectroscopic analysis of all the derivatives of newly synthesized thiourea was carried out. ^1^H and ^13^C NMR were recorded in deuterated DMSO-d6 solvent. The products’ constructions were supported by their ^1^H NMR and ^13^C NMR spectrum (Experimental detail about characterization data; FTIR, ^1^H and ^13^C NMR spectra are depicted in Appendix A provided in Appendix A). In ^1^H NMR, two N-H protons as singlets appeared at 12.161 ppm and 11.815 ppm and were a clear indication of thiourea formation. Intramolecular hydrogen bonding shifted these signals to a higher ppm value; hence the thio core in the thiourea structure is justified. The 8.037–7.290 ppm region indicates the aromatic rings. The most shielded signal in all the structures is of the methyl group attached to the aromatic ring which appeared in the region of 2–2.5 ppm. In ^13^C NMR, strong signals at 180 ppm and in the range of 175–160 ppm are a clear indication of C=S and C=O groups. The signal for C=S carbon is the de-shielded one which appeared at 181–180 ppm while the signal for C=O appeared at 175–160 ppm. Signals between 120 and 140 ppm represent aromatic carbons and the signal for methyl carbon which is directly connected to the aromatic ring appeared in the region of 6.8–6.2 ppm. In FTIR, a broad band above 3200 cm^−1^ shows NH stretching because of the intra-molecular hydrogen bond between the NH and carbonyl oxygen around 3000 cm^−1^. The C=O stretch resulted in the appearance of an intense band in the region of 1700–1600 cm^−1^; the signal in the region of 1050–1250 cm^−1^ was a clear indication of C=S for all the compounds.

### 2.3. Biological Activity

#### 2.3.1. Free Radical Scavenging

All of the produced 1-aroyl-3-[3-chloro-2-methylphenyl]thiourea compounds were assessed for the ability of DPPH free radical scavenging activity as depicted in Figure 2.

The anti-oxidant activity suggested that compounds **4d**, **4e**, **4f**, **4g**, **4h** and **4i** showed good activity in the comparison of standard Vitamin C. However, the rest of the compounds did not display substantial radical scavenging activity even at the high concentration [100 µg/mL].

#### 2.3.2. In Vitro Urease Inhibitory Activity

In the present work, 1-aroyl-3-[3-chloro-2-methylphenyl]thiourea hybrids [**4a**–**j**] were synthesized with the aim of having potent JBU inhibitors. Hydrophobic and hydrophilic groups were substituted on a phenyl ring in the novel analogues **4a**–**j** that were synthesized in order to check urease inhibition activity.

The anti-urease activity results depicted the influence of different functional groups on enzyme activity. The function of nitro, chloro, methyl and ethyl substitution in urease inhibitory activity was evaluated. Thiourea, a well-known urease inhibitor, served as the reference compound. All of the synthesised compounds demonstrated good to exceptional urease inhibitory efficacy relative to the reference medication (thiourea). The IC_50_ ranged from 0.0019 ± 0.0011 to 0.0532 ± 0.9951 µM and are far better than standard (thiourea) with IC_50_ 4.7455 ± 0.0545 µM (Table 1). A total of ten derivatives of N-[[3-chloro-2-methylphenyl]carbamothioyl]benzamide (**4a**–**j**), possessed different electron-withdrawing and -donating substituents on the benzamide ring; methoxy [**4b** & **4c**], methyl (**4d**), nitro (**4e**, **4f** & **4g**) and chloro (**4h**, **4i** & **4j**) groups.

### 2.4. Structure Activity Relationship (SAR)

Briefly, when the SAR of compounds **4b** and **4c** was compared with the parent compound **4a**, it was observed that the induction of the methoxy group resulted in improved activity. The behaviour of methoxy in position 4 (para position) has a better inhibitory effect than in position 3, 5 (meta position). In the para position the methoxy group acted as an electron-donating group with a resonance phenomenon while in the meta position it behaved as an electron-withdrawing group by inductive effect. When the effect of the nitro group was noticed, the derivative 4e exhibited maximum inhibitory activity in comparison to 4f and 4g. When the SAR of this derivative 4e was detected, it was found that para substitution resulted in enhanced inhibitory potential as compared to meta directing substitutions. It was also noted that induction of one NO*_2_* group in the meta position resulted in better activity [IC_50_ ± SEM = 0.0136 ± 0.0544] as compared to the derivative 4g with an inhibitory value of IC_50_ ± SEM = 0.0335 ± 0.0994. This paper reveals that the deactivating nature of the nitro group exceeds the activating nature of the methyl group in such a way that the SAR of derivative 4d indicated that methyl substitution in position 4 (para position) was not strong enough to improve the inhibition as compared to the parent compound (**4a**). Interesting behaviour was seen when the substitution of the strong electronegative functional group was introduced to the benzamide ring. Further investigation suggested that the induction of the electronegative group in position 2 (ortho) resulted in improved inhibitory potential in comparison to **4h** and **4j**. Among all the studied compounds, the derivative **4i** was discovered to be the most effective inhibitor. The derivative **4j** having ortho & para chloro substitutions had less activity as compared to **4i** but more than **4h**. When the SAR of **4h**, **4i** & **4j** was compared, it was observed that substitution of the ortho position is more favourable as compared to para (**4h**) and even di-substitution at ortho & para (**4j**). From the SAR it can be concluded that para-directing substitutions have better inhibitory activities owing to the fact that we already know that para and ortho positions for substitutions are more stable and faster to prepare. In the same manner, halogens showed their behaviour, as chlorine is more electronegative and when it is attached in the ortho position (**4i**), it gives a more potent inhibitory effect than in the para position (**4h**) and ortho, para position (**4j**). Based upon our results, for the discovery and development of new urease inhibitors, **4i** can be used as a structural model.

### 2.5. Kinetic Analysis

Kinetic investigations provided additional evidence for the inhibitory function, in which the potential of the potent derivative **4i** to inhibit the substrate at different concentrations in such a way that enzyme concentration remained constant. The kinetic tests of the enzyme gave a series of straight lines by the Lineweaver–Burk plot of 1/V versus 1/[S] in the presence of varying compound concentrations (Figure 3A). The results showed that the compound **4i** intersected in the second quadrant. Thus, the analysis revealed that V_max_ declined to new growing doses of inhibitors; however, Km remained the same. This behaviour revealed that the mode of enzyme-inhibitor complex shown by compound **4i** inhibition is non-competitive. The second plot of the slope against the inhibitor‘s concentration showed a dissociation constant (Ki) of enzyme inhibition (Figure 3B).

We selected the most effective compound **4i**, based on our results, to evaluate its type of inhibition and inhibition constant on JBU (Figure 3). Table 2 shows the kinetic results.

### 2.6. Computational Studies

#### 2.6.1. Jack Bean Urease Structural Analysis

JBU comprises four domains with two nickel atoms, each with diverse numbers of amino acids (Figure 4). Detail structural analysis revealed that in domain 4, metal binding residue (His545, His519, His409, His407 and Asp633) nickel atoms showed direct interactions within the active pocket of JBU. The VADAR investigations presented that the protein structural design is entailed with helices (27%), β sheets (31%) and coils (41%) in the target protein. Moreover, Ramachandran plots (Appendix A provided in Appendix A) indicated that 97.5% of residues were present in favoured regions that show the precision of angles (phi (φ) and psi (ψ) among the coordinates of JBU.

#### 2.6.2. Binding Pocket Analysis

The binding pocket of JBU shown in Figure 5 had a docked complex of **4i** which was further investigated for the presence of binding interactions. The examination of all the docked complexes was done based on the minimal docking energy values (kJ/mol). Docking results confirmed that compound **4i** presented a strong binding energy value (−28.76 kJ/mol) in comparison to other compounds. The parent compounds **4a** showed the energy value (−26.4 kJ/mol) against urease. Despite the fact that all compounds have the same fundamental nucleus, compounds usually have excellent energy efficiency values and there are not many energy value variances among all designed ligands. The active compound (**4i**) predicted that the ligand binds inside the active region of the targeted protein.

#### 2.6.3. Molecular Docking

To figure out which conformational position of synthesised ligands (**4a**–**j**) is the best fit against JBU docking studies were employed. In detail, the structure activity relationship (SAR) study revealed that two hydrogen bonds and hydrophobic interaction were detected in the **4i**-docking complex at one point. The sulfur group formed a hydrogen bond with Arg439 with a bond length of 2.45 Å (Figure 6A,B). The amino group formed another hydrogen bond interaction with Ala636 having bond distances of 3.25 and 4.37Å, respectively. The relative binding energy and SAR analysis displayed the significance of the **4i** compound and may be considered as effective inhibitors by targeting JBU. Other amino acid residues, i.e., Ala436, Gln635 and Asp494, were involved in van der Waals interactions.

The structure of the ligand is indicated in the olive drab colour; yellow represents the two nickel atoms. Two interacted residues Ala636 and Ala436 are highlighted in purple for hydrogen and hydrophobic bonding, respectively. The 2D depiction of docking complexes of synthesized compounds is shown in the Appendix A.

#### 2.6.4. Molecular Dynamics Simulations

The distinctive feature of CHARMM-GUI is its adaptability, which gives users the choice to select popular simulation programmes for their simulations. Using the NAMD-generated input files, each system was equilibrated for 100 ns production MD simulation. To estimate the stability of the system, root mean square deviations (RMSD) of backbone atoms with respect to the initial structure and root mean square fluctuation (RMSF) of each residue were calculated. For these calculations, the production trajectories were positioned in relation to the atoms of the corresponding segments of protein, nucleic acid, or carbohydrates.

Protein structural alignment is shown by its RMSD. If the simulation has reached equilibrium satisfactorily, the RMSD analysis demonstrates that fluctuations of the simulation around 1–3 are fairly acceptable [24], whereas greater than 3 changes indicate that the protein is going through a significant conformational change. If, at the conclusion of the simulation, the protein’s RMSD is still rising or falling on average, the system has not yet reached equilibrium and the simulation may not have lasted long enough to go through a thorough analysis. Additionally, this suggests that the ligand has clearly spread out beyond its initial binding location.

The Urease-**4i** complex’s RMSD plotted graph (Figure 7) demonstrates that the complex reaches stability at 17 ns. Following that, oscillations in RMSD values for the target (protein) stay within 2.0 for the period of the simulation, which is quite acceptable. After being equilibrated, protein-ligand complex RMSD values oscillate within 2.5 angstroms. These findings demonstrate that during the simulated time, the ligand remained firmly bound to the receptor’s binding site.

Peaks in the RMSF show residues of the protein that fluctuate during the simulation. Usually, the tail (N and C terminal) of the protein changes more than other parts. The residues with higher peaks are found in the N and C terminal zones, according to MD trajectories (Figure 8). The stability of the ligand–protein interaction is shown by the low RMSF values of the binding site residues.

The radius of gyration (rGyr) refers to the compactness and distribution of amino acid residues around the center of mass. The high value for rGyr indicated that protein has a low stability and uneven distribution of residues. In the current study, amino acid residues showed consistent value for the radius of gyration which ranges from 30–31 angstroms. An MD simulated trajectory showed slight fluctuation at 60 and 80 ns but it became stable after a short period of time. Figure 9 shows the rGyr value for the Urease-**4i** complex.

The solvent accessible surface area (SASA) corresponds to the exposure of the surface area of the protein to the solvent. The higher the value of the SASA the lower will be the stability of the protein. In the current study, the residue-based SASA value was retrieved from simulated trajectories. It was observed that the SASA value for amino acid residues of the targeted protein ranged from 200 to 350 Å^2^. In particular, amino acid 780 showed a slightly higher value for the SASA (Figure 10).

#### 2.6.5. Density Functional Theory (DFT)

The in vitro effect of these derivatives on the highest occupied molecular orbital (HOMO) and lowest unoccupied molecular orbital (LUMO) reflected the biological activity, chemical reactivity and stability of a molecule. A molecule with small frontier orbitals gap revealed high chemical reactivity and low kinetic stability. According to Table 3, the compound which showed the lowest energy gap was compound **4c** (∆E = 0.058 eV) and this was found to be the softest molecule. The compound with the highest energy gap was the compound **4g** (∆E = 0.114 eV) and it was predicted to be the most kinetically stable of all the compounds. As HOMO is the electron donor and LUMO is the electron acceptor, the compound having the highest HOMO energy was **4c** (EHOMO = −0.097 eV). It was thought to be the best electron donor due to its higher energy. The compound with the lowest LUMO energy was the compound **4c** (ELUMO = −0.039 eV), which suggests that it can be the best electron acceptor. As the compound which has small orbital energy gap is more polarized, compound **4c** was identified as being more polarized. The geometrical optimization and HOMO−LUMO is given in Figure 11 and Figure 12 and values are given in Table 3.

#### 2.6.6. ADMET Properties

Ten derivatives of thiourea were analyzed for their ADMET properties. It was established that all the compounds met the criteria of Lipinski’s Rule of Five (Table 4).

All molecules were shown to have an optimal volume of distribution range. Furthermore, it was discovered that almost all derivatives had superior CACO-2 permeability to ordinary thiourea, with the exception of compound **3**, which had a negative value of caco-2 permeability. The computed value of human intestinal absorption of all substances was found to have a high chance of being HIA+. The HIA+ of all compounds was comparable to that of conventional thiourea. The greater the HIA number, the better the compound’s intestinal absorption. A chemical with a positive blood brain barrier value has a better lipophilicity profile and may quickly absorb from plasma membranes; the computed blood brain barrier value for all compounds was found to be closer to standard thiourea, which was BBB+ in all compounds. In terms of the PGP substrate, according to standard thiourea, the output value of all compounds had a high likelihood of being aPGP substrate. Furthermore, ordinary thiourea is a powerful inhibitor of P-glycoproteins, and all derivatives were found to be effective inhibitors of P-glycoproteins. In general, all compounds had a superior ADMET profile to the standard; all values are listed in Table 5.

## 3. Materials and Methods

Prior to use, standard methods were followed for the drying and distillation of solvents. Sigma Aldrich provided all the reagents. A Stuart SMP3 melting point apparatus was used for determining the melting point. The NMR spectra were recorded on a Bruker 300 (^1^H-NMR at 300MHz and for ^13^C-NMR at 75.5MHz) and chemical shifts are stated in ppm against an internal reference standard tetramethylsilane or the residual solvent resonance [25]. A thin layer chromatography (TLC) was used to monitor the reactions, and aluminum sheets coated with silica gel F254 were used (Merck). Ultravoilet light at 254 and 360 nm was used for the detection of invisible spots on the TLC plate.

### 3.1. General Procedure

All the chemicals such as aromatic acids, aniline [2-Me-3-Chloroaniline] and thionyl chloride were bought from Aldrich. Acetone that was of analytical grade [E. Merck] was dried and distilled freshly. One mmol of each aromatic acid was kept in a 100 mL of two roundneck bottom flasks, attached to a reflux condenser and a gas trap. In 1.2 mmol thionyl chloride a few drops of dry DMF were added and the reaction mixture was refluxed for 3 h in order to obtain the acid chlorides. In a 250 mL of two roundneck bottom flasks attached to a flux condenser, a solution of potassium thiocyanate (1 mmol) in 20 mL dry acetone was stirred; freshly manufactured aromatic acid chlorides (1 mmol) in each case were supplied dropwise and the mixture was refluxed for 30 min with the addition of the solution of 2-Me-3-chloroaniline (1 mmol) in 20 mL dry acetone dropwise. The reaction mixture was then refluxed and stirred for 1–2 h. TLC [thin layer chromatography] was employed to monitor the progression of the reaction. After the reaction was complete, the reaction mixture was dumped onto crushed ice. The solid thiourea precipitates that appeared instantly were then filtered out, cold distilled water was used for washing it well; it was dried and recrystallization was completed in ethanol to synthesize thiourea derivatives **4a**–**j** (Figure 1).

### 3.2. Biological Activities

#### 3.2.1. Free Radical Scavenging Assay

The assay performed in our studies had the same methodology as reported in our previous research articles [25,26].

#### 3.2.2. In Vitro Urease Inhibitory Activity

The methodology used in this research for in vitro urease inhibitory activity is the same as reported in our earlier paper [27].

#### 3.2.3. Kinetic Analysis

Kinetics performed here as stated previously in our research articles was used for urease activity determination [25,26,27]. For this analysis, compound **4i** having the most potent IC_50_ value was selected.

### 3.3. Computational Studies

#### 3.3.1. Retrieval of Jack Bean Urease

The crystal structure of JBU was downloaded from the Protein Data Bank (PDB) [28] with PDBID4H9M on the basis of high resolution. The UCSF Chimera 1.10.1 was utilized to access the protein structure of JBU [29]. The Ramachandran graph of the targeted protein was obtained by employing the Discovery Studio Visualizer 4.1 [30]. For the prediction of protein architecture statistical percentage values of the receptor proteins helices, β-sheets, coils and turn, an online server VADAR 1.8 was used [31].

#### 3.3.2. Preparation of Ligands and Molecular Docking

All the synthesized chemical ligands were drawn with an ACD/ChemSketch tool and made smaller with an UCSF Chimera 1.10.1 tool. All the ligands [**4a**–**j**] were docked against the crystal structure of the JBU experimentally by using a PyRx docking tool [32]. In order to visualize the binding conformational analysis, a grid box center with dimensions 60, 60 and 60, for X, Y and Z, respectively, with spacing of 0.375 Å, was tuned by default exhaustiveness value, which is 8. The evaluation of all the docked complexes was carried out on the basis of the values of the lowest binding energy [kJ/mol] and analysis of the structure activity relationship [SAR]. The Discovery Studio Visualizer [4.1] was used to accomplish 3D graphical representation of all the docked complexes.

#### 3.3.3. Molecular Dynamics Simulations

For evaluating a set of optimal simulation protocols in NAMD [33], the first of the entire complex (protein urease and ligand **4i**) was built using CHARM-GUI solution builder. The Monte Carlo method was used for computing the equilibrium properties. Protein was solvated using the TIP3P water model both individually and in complex, then counter NaCl ions were added with 0.15 concentrations to neutralize charges. Initially, NVT equilibration was performed with the temperature set at 300 K and a constant volume throughout one nanosecond [34]. Secondly, NPT equilibration was conducted for one nanosecond with the temperature set at 300 K and a constant pressure [35]. Ten thousand frames of each trajectory were captured while simulating both proteins and their complexes at 100 ns. Periodic boundary conditions (PBC) were set up automated. CHARMM36 were selected as a forcefield here. VMD was utilized to analyze the results [36].

#### 3.3.4. Density Functional Theory

The density functional theory is the wonderful technique used to obtain the equitable optimized structures and the HOMO LUMO analysis. B3LYP functional was used for great accuracy and the efficiency of vibrational spectra. All the gas-phase calculations performed by the DFT/B3LYP method with STO-3G basis set used the Gaussian 09 program [37]. Results were obtained by visualizing the output files using the Gauss View 5 program [38].

#### 3.3.5. ADMET Properties

Predicting ADMET characteristics is a crucial step in the drug development process. Drug design and lead optimization are aided by in silico ADMET evaluation models. The online web server ADMET lab 2.0 [ADMET lab 2.0 https://admetmesh.scbdd.com/ (accessed on 7 July 2022) was used to calculate the physicochemical parameters and medicinal properties of thiourea derivatives [39].

## 4. Conclusions

The derivatives 1-aroyl-3-[3-chloro-2-methylphenyl]thiourea hybrids **4a**–**j** were efficiently synthesized with a high yield. The chemical structures of the synthesized compounds were characterized by spectral data such as FTIR, ^1^H and ^13^C NMR. In-vitro results of enzyme inhibitory activity proved that the compound **4i** exhibited exceptional enzyme inhibitory activity with IC_50_ value 0.0019 ± 0.0011 µM better than standard thiourea (4.7455 ± 0.0545 µM). The kinetic studies were also carried out and all the synthesized compounds [**4a**–**4j**] showed effective anti-urease activity. Moreover, in silico investigations including molecular docking studies, density functional theory (DFT) studies, and ADMET properties further supported these studies. Therefore, it is suggested that 1-aroyl-3-[3-chloro-2-methylphenyl] thiourea derivatives may be the possible “lead” candidate for therapeutic development and discovery.

## Data Availability

Not applicable.

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
