# Peer review of "Analysis of 1-Aroyl-3-[3-chloro-2-methylphenyl] Thiourea Hybrids as Potent Urease Inhibitors: Synthesis, Biochemical Evaluation and Computational Approach"

_ijms, 2022, doi:10.3390/ijms231911646_

Round 1

Reviewer 1 Report

Comments for the authors

1. All the abbreviations in the manuscript should be checked.

2. Please align captions of the tables.

3. Table 4 and 5 are not in the scientific format.

4. First line in the conclusion should be rephrased to “The derivatives 1-aroyl-3-(3-chloro-2 methylphenyl)thiourea hybrids 4a-j have been 430 efficiently synthesized with high yield”.

5. Please replace word “Products” with “compounds” in the third line of conclusion

6. The term in silico must be in italic form “in silico”

7. Please rephrase the last sentence in the conclusion to “Therefore it is suggested that of 1-aroyl-3-(3-chloro-2-methylphenyl) thiourea derivatives might be the possible "lead" candidates for thera-peutic development and discovery”

8. Please enhance resolution of Figure 2.

9.     Please update the references as per the journal style.

Overall paper is interesting, and can be considered for publication after addressing the minor issues.

Author Response

Comments and Suggestions for Authors

  1. All the abbreviations in the manuscript should be checked.

Response: Abbreviations and formatting related issues have been omitted as suggested.

  1. Please align captions of the tables.

Response: All table captions have been aligned.

  1. Table 4 and 5 are not in the scientific format.

Response: Both tables has been modified as per others format.

      4. First line in the conclusion should be rephrased to “The derivatives 1-aroyl-3-(3-chloro-2 methylphenyl)thiourea hybrids 4a-j have been 430 efficiently synthesized with high yield”.

Response: correction has been made as suggested.

      5.Please replace word “Products” with “compounds” in the third line of conclusion

Response: The word product has been replaced with compounds.

  1. The term in silico must be in italic form “in silico”

Response: We have modified the said word.

  1. Please rephrase the last sentence in the conclusion to “Therefore it is suggested that of 1-aroyl-3-(3-chloro-2-methylphenyl) thiourea derivatives might be the possible "lead" candidates for thera-peutic development and discovery”

Response: Changes have been made as suggested.

  1. Please enhance resolution of Figure 2.

Response: Figure 2 resolution has been enhanced.

  1. Please update the references as per the journal style.

Response: References have been updated as per journal style.

Overall paper is interesting, and can be considered for publication after addressing the minor issues.

Thankyou anonymous reviewer for acknowledgment.

Reviewer 2 Report

In this work, the authors synthesize a set of 1-aroyl-3-(3-chloro-2-methlphenyl) thiourea hybrids and use an array of experimental and computational methods to evaluate their characteristics as potential urease inhibitors.

Based on the data obtained, the authors conclude that the thiourea hybrids investigated in the present study has exceptional urease inhibitory activity and suggest that it can be used as a starting point for future research.

Overall, I find the manuscript is clearly written and the techniques employed are comprehensive. Please see the following comments:

Main comments:

1. The IC50 values for the compounds investigated are provided in Table 1. My question is why the values of SEM (standard error of mean) are larger than the mean values. For instance, for compound 4d, IC50=0.0532(+/-)0.9951. SEM here is almost 20 times its mean value. This means that there is a very large uncertainty associated with the IC50 value. Thus casting doubt on the basic conclusion of the present study.

2. The authors computed RMSD (Figure 7) to show that during the simulation “the ligand remains firmly bound to the receptor’s binding site”. RMSD is calculated by averaging over all the atoms from both protein and ligand. Since the number of atoms of protein is much higher than that of ligand, the value of RMSD is mostly contributed by the protein. Hence, RMSD may not be sensitive to the motion of the ligand relative to the receptor. I suggest that the authors calculate a quantity that directly measures the binding of the ligand to the receptor (for instance, the center of the mass distance between the ligand and the binding site) to demonstrate that the ligand bind firmly to the binding site over the course of the simulation.

Other comments:

  1. Section 2.2 Spectroscopic characterization. I suggest that the authors provide figures in this section. For instance, the figures for NMR results should be provided either in the main text or in the SI and should be properly referred to in this section.
  2. Section 2.3.1 Free radical scavenging. On page 4, top of the page, the authors claim that “rest compounds did not display substantial radical scavenging activity even at the high concentration (100 \mu g/mL)”. Can the author provide a figure or some data to support this claim? And also the concentration corresponding to Figure 2 should be stated in the text.
  3. On page 8, the authors claim that “RMSD analysis demonstrates fluctuations of the simulation around 1-3 are fairly acceptable”, can the authors provide support or references to the number “1-3”?
  4. In Figure 8, I suggest that the authors use color to mark the residues of the binding site.
  5. In Figure 12, what do green and red colors indicate, respectively?
  6. On page 6, the first sentence in section 2.6.1, “ywo” should be “two”
  7. Figure 3b is cropped. A bigger and not cropped version should be provided.

Author Response

Comments and Suggestions for Authors

In this work, the authors synthesize a set of 1-aroyl-3-(3-chloro-2-methlphenyl) thiourea hybrids and use an array of experimental and computational methods to evaluate their characteristics as potential urease inhibitors.

Based on the data obtained, the authors conclude that the thiourea hybrids investigated in the present study has exceptional urease inhibitory activity and suggest that it can be used as a starting point for future research.

Overall, I find the manuscript is clearly written and the techniques employed are comprehensive. Please see the following comments:

Main comments:

  1. The IC50 values for the compounds investigated are provided in Table 1. My question is why the values of SEM (standard error of the mean) are larger than the mean values. For instance, for compound 4d, IC50=0.0532(+/-) 0.9951. SEM here is almost 20 times its mean value. This means that there is a very large uncertainty associated with the IC50 value. Thus casting doubt on the basic conclusion of the present study.

Response: Thank you for your comment, the standard error of the mean has been calculated for the repeated experiments differences. For compound 4d yes there was a typo mistake it is actually the (SEM= 0.0995 µM). But a little higher SEM can be possible because it has been calculated from the errors of the repeated experiment. For this compound, we have corrected it in the manuscript file. A little higher SEM can be possible the reference as followed;

Mustafa, Muhammad Naeem, et al. "Synthesis, molecular docking and kinetic studies of novel quinolinyl based acyl thioureas as mushroom tyrosinase inhibitors and free radical scavengers." Bioorganic Chemistry 90 (2019): 103063.

  1. The authors computed RMSD (Figure 7) to show that during the simulation “the ligand remains firmly bound to the receptor’s binding site”. RMSD is calculated by averaging over all the atoms from both protein and ligand. Since the number of atoms of protein is much higher than that of ligand, the value of RMSD is mostly contributed by the protein. Hence, RMSD may not be sensitive to the motion of the ligand relative to the receptor. I suggest that the authors calculate a quantity that directly measures the binding of the ligand to the receptor (for instance, the center of the mass distance between the ligand and the binding site) to demonstrate that the ligand bind firmly to the binding site over the course of the simulation.

Response: Yes author is right, in protein-ligand complex RMSD is majorly contributed by protein, so for taking into account the RMSD of ligand, we have calculated RMSD value for ligands atoms only and it was observed that ligand RMSD remained below 1.5 angstrom. This RMSD pattern depicts the optimal deviation of ligand inside the active pocket of protein. Figure is presented below. Moreover, rGyr is also presented in the manuscript which also affirm the compactness and stability of complex.

Section 2.2 Spectroscopic characterization. I suggest that the authors provide figures in this section. For instance, the figures for NMR results should be provided either in the main text or in the SI and should be properly referred to in this section.

Response: Spectra’s have been provided in supplementary file and citation has been made in main text.

Section 2.3.1 Free radical scavenging. On page 4, top of the page, the authors claim that “rest compounds did not display substantial radical scavenging activity even at the high concentration (100 \mu g/mL)”. Can the author provide a figure or some data to support this claim? And also the concentration corresponding to Figure 2 should be stated in the text.

Response: The concentration which has been used is mentioned in the text in the same section (2.3.1) at the end of paragraph, and the % activity showed by compounds are presented in figure 2.

On page 8, the authors claim that “RMSD analysis demonstrates fluctuations of the simulation around 1-3 are fairly acceptable”, can the authors provide support or references to the number “1-3”?

Response: Reference (reference # 24) has been added as per suggestion.

In Figure 8, I suggest that the authors use color to mark the residues of the binding site.

Response: Binding site residues have been marked as suggested.

In Figure 12, what do green and red colors indicate, respectively?

Response: Red, yellow and blue atoms indicate C, S and O atoms. Deep-red and deep-green parts indicates to the different phases of molecular wave functions. Same has been incorporated in the main text.

On page 6, the first sentence in section 2.6.1, “ywo” should be “two”

Response: Thank you for your deep checking and notice, we have corrected the typo error.

Figure 3b is cropped. A bigger and not cropped version should be provided.

Ans: This Figure is combined as individual figures in the one, however, we have modified and added it in the manuscript file.

Reviewer 3 Report

The authors have described the synthesis of thiourea based hybrids as potent urease inhibitors and a computational approach. The manuscript's content is acceptable for publishing in IJMS. However, additional comments are required prior to potential publication. 

1.

Abstract: Please make a bold for compounds 4i and check throughout the manuscript. As well (line no # 32), in “IC50” 50 should be subscript.

2.

Introduction (Line # 71; Figure 1): Remove the “Already”

3.

The authors refer to the following articles describing the importance of molecular hybridization in developing hybrid molecules in drug discovery. a). Medicinal Chemistry Research (2022) 31:1088–1098 (https://doi.org/10.1007/s00044-021-02835-1)

b.     European Journal of Medicinal Chemistry 188 (2020) 111974 (https://doi.org/10.1016/j.ejmech.2019.111974)

c.      European Journal of Medicinal Chemistry 114 (2016) 293-307 (http://dx.doi.org/10.1016/j.ejmech.2016.03.013)

4.

Synthesis: The authors only focused on synthesizing various acid-based derivatives and kept aniline consistent; what is the significant reason behind this?

5.

How stable is the intermediate? And what are the essential structural features of activity?

6.

References: In the text, reference numbers should be placed in square brackets [], and references are not in the journal format. Please follow the journal guidelines.

7.

NMR spectra are missed, and it would be good if they included the spectra.

Author Response

Comments and Suggestions for Authors

The authors have described the synthesis of thiourea based hybrids as potent urease inhibitors and a computational approach. The manuscript's content is acceptable for publishing in IJMS. However, additional comments are required prior to potential publication.

  1. Abstract: Please make a bold for compounds 4i and check throughout the manuscript. As well (line no # 32), in “IC50” 50 should be subscript.

Response: The correction has been made as per suggestion of reviewer.

2 .Introduction (Line # 71; Figure 1): Remove the “Already”

Response: mentioned word has been replaced as suggested.

  1. The authors refer to the following articles describing the importance of molecular hybridization in developing hybrid molecules in drug discovery. a). Medicinal Chemistry Research (2022) 31:1088–1098 (https://doi.org/10.1007/s00044-021-02835-1)

b.European Journal of Medicinal Chemistry 188 (2020) 111974 (https://doi.org/10.1016/j.ejmech.2019.111974)

c.European Journal of Medicinal Chemistry 114 (2016) 293-307 (http://dx.doi.org/10.1016/j.ejmech.2016.03.013)

Response: Thank you for suggesting impactful studies, we have referred to these studies and relevant citations have been made.

  1. Synthesis: The authors only focused on synthesizing various acid-based derivatives and kept aniline consistent; what is the significant reason behind this?

Response: Reviewer is right as we have adopted this approach in order to investigate impact of different substituents and structures on the biological activity. So in the current study, the aryl substituents in the aryl part of thiourea were varied which had significant impact on activities. Reference is cited below.

Zaher DM, El‐Gamal MI, Omar HA, Aljareh SN, Al-Shamma SA, Ali AJ. et al. Recent advances with alkaline phosphatase isoenzymes and their inhibitors. Arch Pharm. 2020;353:e2000011

  1. How stable is the intermediate? And what are the essential structural features of activity?

Response: Due to the resonance of the nitrogen ion pair, it is stable in basic environments. The objective of the pharmacophore coordination technique is to produce hybrids with enhanced affinities and potencies by fusing at least two pharmacophore moieties from distinct bioactive particles into a structural unit. In addition, it can lead to compounds with enhanced selectivity profiles, dual modes of action, and diminished side effects. (Fig.).

  1. References: In the text, reference numbers should be placed in square brackets [], and references are not in the journal format. Please follow the journal guidelines.

Response: All references are converted to journal format.

  1. NMR spectra are missed, and it would be good if they included the spectra.

Response: All available spectra’s have been included in supplementary file.
